# The m15 Locus of Murine Cytomegalovirus Modulates Natural Killer Cell Responses to Promote Dissemination to the Salivary Glands and Viral Shedding

**DOI:** 10.3390/pathogens10070866

**Published:** 2021-07-09

**Authors:** Baca Chan, Maja Arapović, Laura L. Masters, Francois Rwandamuiye, Stipan Jonjić, Lee M. Smith, Alec J. Redwood

**Affiliations:** 1School of Biomedical Sciences, University of Western Australia, Crawley, WA 6009, Australia; baca.chan@uwa.edu.au (B.C.); laura.masters@uwa.edu.au (L.L.M.); Francois.Rwandamuriye@telethonkids.org.au (F.R.); lees@speedx.com.au (L.M.S.); 2Institute of Respiratory Health, University of Western Australia, Nedlands, WA 6009, Australia; 3Department for Histology and Embryology, Faculty of Medicine, University of Rijeka, 51000 Rijeka, Croatia; maja.arapovic@mef.sum.ba (M.A.); stipan.jonjic@medri.uniri.hr (S.J.)

**Keywords:** cytomegalovirus, immune evasion, m15, natural killer, coterminal transcription, salivary gland, saliva shedding

## Abstract

As the largest herpesviruses, the 230 kb genomes of cytomegaloviruses (CMVs) have increased our understanding of host immunity and viral escape mechanisms, although many of the annotated genes remain as yet uncharacterised. Here we identify the m15 locus of murine CMV (MCMV) as a viral modulator of natural killer (NK) cell immunity. We show that, rather than discrete transcripts from the m14, m15 and m16 genes as annotated, there are five 3′-coterminal transcripts expressed over this region, all utilising a consensus polyA tail at the end of the m16 gene. Functional inactivation of any one of these genes had no measurable impact on viral replication. However, disruption of all five transcripts led to significantly attenuated dissemination to, and replication in, the salivary glands of multiple strains of mice, but normal growth during acute infection. Disruption of the m15 locus was associated with heightened NK cell responses, including enhanced proliferation and IFNγ production. Depletion of NK cells, but not T cells, rescued salivary gland replication and viral shedding. These data demonstrate the identification of multiple transcripts expressed by a single locus which modulate, perhaps in a concerted fashion, the function of anti-viral NK cells.

## 1. Introduction

Cytomegaloviruses (CMVs) are large double-stranded DNA viruses and members of the betaherpesvirus subfamily of the herpesvirus family. These viruses are highly species specific and have co-evolved with their respective hosts for millions of years [1]. CMVs have developed a sophisticated array of mechanisms to subvert host defences and aid in the establishment of persistent, life-long infections. Murine CMV (MCMV) encodes multiple genes dedicated to antagonising natural killer (NK) cell responses, reflective of the critical role these cells play in host defence against MCMV infection [2,3]. MCMV modulation of NK cells is highly effective as most laboratory mouse strains are unable to control acute MCMV replication in the spleen and liver [4].

The m02 and m145 gene families, located at either termini of the genome, are unique to MCMV and dispensable for in vitro replication [5,6]. Many of the known NK immunomodulatory genes are contained within these two MCMV gene families (reviewed [7,8]). MCMV employs a range of NK evasion strategies, exemplified by the downregulation of MHC class I molecules and the stress-inducible ligands RAE-1, MULT-1 and H60 which bind to the NK cell activation ligand NKG2D. The absence of these NK evasion genes generally results in defective viral replication within the host. Nonetheless, there is strong evidence that there remains other, as yet unidentified, MCMV genes which target NK cells. In a recent study demonstrating that the m20.1 protein retains CD155/PVR, the ligand of activation receptor DNAM-1, in the ER leading to its degradation [9], it was observed that an unknown viral protein interacts with a different DNAM-1 ligand, nectin-2. Likewise, in the characterisation of the m154 protein, which downregulates surface expression of CD48 on infected macrophages to inhibit NK cell responses, transfection of m154 alone was insufficient to rescue the phenotype [10,11].

The identification of MCMV encoded immunomodulatory genes has most typically involved the study of one of two laboratory strains of MCMV, Smith or K181. However, MCMV demonstrates considerable sequence variation within the immunomodulatory genes [12,13,14]. Sequence variation between the viral strains can have significant effects on in vivo replication, across a range of inbred mouse strains [13]. Differences in in vivo replication is due, at least in part, to the expression of the particular genotypes (or alleles) of MCMV immunomodulatory genes combined with the specific alleles of host target proteins [15]. For instance, the most well characterised member of the m145 gene family, m157, exists as different genotypes [13,16]. The genotype expressed by Smith and K181 binds the NK activation ligand, Ly49H, leading to NK cell mediated control [17,18]. However, most m157 genotypes do not engage Ly49H [15]. Conversely, expression of Ly49H is rare in inbred mouse strains, for example restricted to C57BL/6 and Ma/My mice, and therefore it has been proposed that m157 likely evolved to target inhibitory Ly49 members consistent with a role in immune subversion. Indeed, Smith genotype m157 engages inhibitory ligand Ly49I, but not activation ligand Ly49U, expressed on NK cells in susceptible 129/J mice [19]. Variants of m157 from wild MCMV strains engage a range of inhibitory Ly49 receptors in BALB/c, C57BL/6 and NZB mice [15]. Sequence variation in the host and the virus therefore has the potential to complicate gene function analyses. Another factor potentially confounding traditional gene knockout studies is the apparent complex transcriptional regulation evident in CMV [20,21].

In this study, we sought to characterise m15, one of the most variable genes in MCMV [13]. To account for the potential confounding effects of sequence variation we assessed viral replication in multiple genetically distinct mouse strains. Deletion within the 3′-terminus of the m15 gene led to salivary gland attenuation in all strains of inbred mice tested, however a larger deletion at the 5′-terminus had no effect, nor did single nucleotide insertions introducing premature stop codons in the m14, m15 or m16 gene. Transcriptional mapping of the m15 locus identified five 3′-coterminal transcripts spanning m14 to m16. Attenuation in the salivary glands required the disruption of all five transcripts. In the absence of these transcripts, acute NK cell responses in MCMV infected mice were heightened with evidence of prolonged activation, elevated IFNγ production and enhanced proliferation. This study identifies the m15 locus as a novel inhibitor of NK cells, leading to the spread of MCMV to the salivary glands and shedding into the saliva from this key site.

## 2. Results

### 2.1. Disruption of m15 Attenuates Salivary Gland Replication in Multiple Mouse Strains

It is well established that susceptibility to MCMV in mice depends on both host and viral genetics. Our interest in the m15 gene initially arose from our sequencing studies identifying it as one of the most genetically variable regions within the MCMV genome [13], suggesting that it would interact with a highly variable host factor that may not be present in all mouse strains. To assess the effect of m15 on viral replication, we deleted 110 bp from the 3′ region of m15, avoiding the overlap with the m16 gene (MCMV-∆m15; Figure 1A). MCMV-∆m15 demonstrated no growth defect in fibroblasts (Appendix A). In the first instance we assessed the replication of MCMV-∆m15 compared to a revertant virus, MCMV-Rev, in BALB/c mice. MCMV-Rev replicates like wild type MCMV in vitro and in vivo (Appendix A). In the spleen of infected BALB/c mice, MCMV-Δm15 titres were significantly elevated compared to MCMV-Rev titres (Figure 1B). A similar, but not significant, trend of enhanced replication for MCMV-Δm15 was likewise observed in the liver. In contrast to elevated titres seen during acute infection, MCMV-Δm15 was highly attenuated during chronic infection at day 18 post-infection in the salivary glands (Figure 1C). This difference was not due to adventitious mutations within MCMV-Δm15 because MCMV-rev retained wild type replication kinetics (Appendix A).

We additionally assessed the replication of MCMV-Δm15 in inbred mouse strains SJL and CBA to extend the analysis to hosts with diverse genetic backgrounds and resistance to MCMV [22,23]. Due to the higher resistance of SJL and CBA mice to MCMV compared to BALB/c mice [22], acute infection in SJL and CBA mice was assessed using salivary gland-derived virus (SGV), which is more virulent than the equivalent tissue culture-derived virus (TCV). In acutely infected SJL mice, MCMV-Δm15 replicated to similar levels to MCMV-Rev in both the spleen and liver (Figure 1D). MCMV-Δm15 titres were slightly, but not significantly, higher than MCMV-Rev in the spleen of CBA mice, whereas no differences in titres were observed in the liver (Figure 1F).

We next assessed persistent replication in the salivary glands of SJL and CBA mice. MCMV-Δm15 was significantly attenuated compared to MCMV-Rev in the salivary glands of SJL mice (Figure 1E). In CBA mice, infectious MCMV-Δm15 was not detected in the salivary glands (Figure 1G). Therefore, we observe a consistently positive role for m15 in MCMV replication in the salivary glands of all three strains of mice. Notably, while m15 plays a crucial role in MCMV replication the salivary glands, the degree of attenuation observed for MCMV-Δm15 is host strain dependent.

### 2.2. The 3′-Terminus of m15, but Not the m14, m15 or m16 Genes Are Required for Salivary Gland Replication

The disruption in the m15 gene led to a consistently profound defect in the capacity of MCMV to disseminate to, or replicate within, the salivary glands of three independent mouse strains. In contrast, elevated acute virulence following MCMV-Δm15 infection was not only subtle, but varied between mouse strains and even between experiments (data not shown). Therefore, we focused our studies on understanding the mechanisms and the viral gene/s responsible for MCMV attenuation in the salivary glands. For these experiments, we focused most of our attention on infection studies in CBA mice where the phenotype was most striking.

The 3′-terminus of m15 overlaps the 5′-terminus of the m16 gene by 22 bp. While the 110 bp deletion in the m15 gene in MCMV-Δm15 avoided this overlap, there is the potential that this mutation would affect the 5′-UTR or promoter region of m16. To address this, we constructed a new recombinant virus, with a 358 bp deletion at the 5′-terminus of the m15 gene (Figure 2A). Surprisingly, upon infection of CBA mice, MCMV-5′Δm15 replicated to titres comparable to MCMV-Rev in the salivary glands (Figure 2B). This indicates that while deletion at the 3′-terminus of the m15 gene was detrimental to MCMV replication, a much larger, frameshift inducing, deletion at the 5′-end had no effect on MCMV titres in the salivary glands of mice.

We then took a more refined mutagenesis approach. We disrupted the translation of the m15 gene by introducing a T to A substitution at nucleotide (nt) position 14, 113 (accession #AM886412), resulting in a tyrosine (TAT) to stop (TAA) substitution at the 7th codon of the m15 gene (Figure 2A). MCMV-m15stop also replicated like wild type MCMV in the salivary glands of CBA mice, failing to recapitulate the attenuation of MCMV-Δm15 in this organ (Figure 2C).

These data suggested that we may have indeed disrupted the expression of m16 with our deletion in MCMV-Δm15. To investigate this, we likewise introduced a premature stop codon at the start of the m16 gene with a C to A substitution at nt 15,087, resulting in a tyrosine (TAC) to stop substitution at the 16th codon (Figure 2A). Like MCMV-m15Stop, MCMV-m16Stop did not show a replication defect in the salivary glands (Figure 2C), indicating that the m16 gene is not responsible for the phenotype we observed for MCMV-Δm15. We additionally generated a recombinant virus with a T to A substitution at nt 15, 125, altering the 29th codon of m16 from a leucine (TTA) to a stop codon (TAA). This introduced a stop codon following two other internal start codons in the m16 gene, and also did not affect salivary gland replication (Appendix A). For completeness, we also generated a MCMV-m14Stop recombinant virus by introducing a G to T substitution at nt 13,135, substituting the 18th codon from a glutamine (GAA) to a stop codon (Figure 2A). As MCMV-m14Stop also replicated like wild type MCMV (Figure 2C), we also ruled out m14 as the gene involved in the defective replication of MCMV-Δm15. These data suggest that no one individual annotated gene within this region was responsible for the observed attenuation of MCMV-Δm15 in the salivary glands. That is, neither m14, m15 nor m16 individually affects salivary gland tropism.

### 2.3. The m15 Locus Produces Multiple Coterminal Transcripts

Whole genome transcription studies have yet to demonstrate a consensus transcriptional pattern for the m15 gene [20,24,25]. Therefore, we comprehensively mapped the region from m14 to m16 genes as an adjunct to gene knockout studies. We performed 5′- and 3′- RACE PCR analysis using primers targeting both the forward and reverse strands spanning the m15 gene, as well as the adjacent m14 and m16 genes. A schematic of the primers and transcripts detected from this region is shown in Figure 3. Five capped transcripts were detected, corresponding to transcription start sites at the beginning of m14 (nt 13,069), within m14 (nt 13,213), at the start of m15 (nt 14,066), within m15 (nt 14,418) and at the start of m16 (14,842). Remarkably, 3′-RACE analysis revealed that all five transcripts were 3′ coterminal at nt 15,695 and included the polyA signal downstream of the m16 gene. No transcripts were detected from the complementary DNA strand. These data indicate that multiple, coterminal transcripts are produced over the conventionally annotated m14, m15 and m16 genes, which will henceforth be referred to as the m15 locus.

### 2.4. Deletion of the Entire m15 Locus Attenuates MCMV Replication in the Salivary Glands

Our data thus far has identified multiple overlapping transcripts produced at the m15 locus. Notably, disruption of four of the five transcripts, as in MCMV-5′Δm15, did not affect MCMV replication, whereas disruption of all five coterminal transcripts, as in MCMV-Δm15, led to attenuation in the salivary glands. These data suggest that no single transcript is responsible for normal MCMV replication in the salivary glands. To confirm this, we constructed a final recombinant virus deleting 2,489 bp of sequence (nt 13,134 to 15,622), effectively removing the m14, m15 and m16 genes (MCMV-∆m14–16; Figure 4A). As for MCMV-∆m15, MCMV-∆m14–16 replicated like WT MCMV in the spleen and liver of BALB/c mice at day 3 post-infection (Figure 4B). More importantly, like MCMV-∆m15, MCMV-∆m14–16 titres were statistically significantly reduced compared to wild type MCMV in the salivary glands of BALB/c mice at day 18 post-infection (Figure 4C).

### 2.5. The m15 Locus Regulates NK Cell Responses

Since T cells play a crucial role in the control and clearance of MCMV from the salivary glands [24,25] we reasoned that attenuation of MCMV-Δm15 in this organ was due to a failure of the virus to counteract host T cell responses. To test this, CBA mice, in which the phenotype is most dramatic, were depleted of T cells by a combined i.p. injection of anti-CD4 (GK1.5) and anti-CD8 (YTS169). Control animals were either injected with isotype control antibody (GL121) or PBS. Both immunocompetent and T cell-depleted animals were i.p. infected with 2 × 10^4^ PFU of either MCMV-Δm15 or MCMV-Rev. Salivary glands were harvested 18 days later to assess viral dissemination and replication. The combined results from two independent experiments are shown in Figure 5A. In mice treated with vehicle-only or isotype control, MCMV-Δm15 was undetected in five of eight mice. Based on assigning limit of detection values (100 PFU/g) to these animals, MCMV-Δm15 reached a mean of 900 PFU/g, which was 15-fold less compared to MCMV-Rev titres with a mean of 1.4 × 10^4^ PFU/g. As expected, viral titres in the salivary glands were increased in the absence of T cell control. However, the attenuation of MCMV-Δm15 relative to MCMV-Rev was maintained, with a 50-fold difference between the two viruses in T cell-depleted mice. We therefore concluded that T cells are not the target of the m15 locus.

We next assessed the role of NK cells in the attenuation of MCMV-Δm15. The replication of MCMV-Δm15 was assayed in CBA mice depleted of NK cells by i.p. injection with anti-asialo GM1. Control mice were injected with PBS or left untreated. Both immunocompetent and NK cell-depleted animals were infected i.p. with 2 × 10^4^ PFU of either MCMV-Δm15 or MCMV-Rev. Viral replication was assessed 18 days post-infection in the salivary glands. The combined data from two independent experiments are shown in Figure 5B. As expected, MCMV-Δm15 remained attenuated in NK cell competent mice. However, in NK cell depleted mice, MCMV-Rev and MCMV-Δm15 replicated to similar levels. These data demonstrate that the m15 locus targets NK cells to promote MCMV replication in the salivary glands.

### 2.6. Disruption of the m15 Locus Promotes Enhanced NK Cell Proliferation, Activation and Antiviral Activity

NK cells are not typically associated with control of viral replication in the salivary glands, therefore suggesting that enhanced NK cell control during acute infection prevents dissemination to this site. We sought to characterise the effect of the m15 locus on NK cells by assaying for the expression of the activation marker CD69, the terminal exhaustion marker KLRG1, IFNγ production, proliferation via BrdU incorporation and maturation status by staining for CD11b and CD27.

As expected, expression of CD69 and KLRG1 was low on NK cells from uninfected mice (Figure 6A). Following MCMV infection, the majority of NK cells became activated, with no difference in CD69 expression on NK cells from MCMV-Rev and MCMV-Δm15 infected mice at 1.5 days post-infection (56% vs. 52%). However, a significantly larger proportion of these activated NK cells were terminally exhausted (CD69+KLRG+) in MCMV-Rev compared to MCMV-Δm15 infected mice (59% versus 49%; *p* = 0.0317).

At 3.5 days post-infection, NK cells were highly activated and terminally differentiated in all infected mice. However, while NK cell activation remained high in MCMV-Δm15 infected mice at day 6.5 days post-infection, the number of activated NK cells had reduced by 50% in MCMV-Rev infected mice by this timepoint. This return to levels closer to those of baseline uninfected mice likely reflects control of acute MCMV-Rev infection, whereas the continued activation of NK cells in MCMV-Δm15 infected mice suggests prolonged exposure to MCMV in the spleen.

The antiviral effector function of NK cells was assessed by measuring IFNγ expression at 1.5 and 3.5 days post-infection (Figure 6B). IFNγ production by NK cells was slightly higher in MCMV-Δm15 infected mice compared to MCMV-Rev infected mice at day 1.5 post-infection. However, by day 3.5 post-infection, IFNγ expression was significantly higher in MCMV-Δm15 infected mice than in MCMV-Rev infected mice. Heightened IFNγ expression by NK cells at 3.5 days post-infection was coincident with elevated NK cell proliferation at this time point in MCMV-Δm15 infected mice (Figure 6A). These data indicate that a functional m15 locus correlates with suppressed IFNγ production and NK cell proliferation.

Finally, we measured the maturation of NK cells in infected mice by assessing surface expression of CD27 and CD11b (Figure 6C). NK cells with the greatest effector function are CD11b^hi^CD27^hi^, and in mice infected with MCMV-Rev, 53% of NK cells were CD11b^hi^CD27^hi^ within 1.5 days of infection. In contrast, significantly fewer (46%, *p* = 0.0023) NK cells were CD11b^hi^CD27^hi^ in MCMV-Δm15 infected mice. The percentage of NK cells with a CD11b^hi^CD27^hi^ phenotype was similar in both groups of infected mice at day 3.5 post-infection, and by day 6.5 the majority of NK cells had matured to a CD11b^hi^CD27^lo^ phenotype in both cohorts of mice.

These data indicate that the m15 locus functions to modulate NK cell function throughout acute infection. The m15 locus dampens NK cell responses by reducing NK cell proliferation and IFNγ production. Surprisingly the m15 locus also appears to affect the rate of NK cell maturation as mice infected with MCMV strains with an intact m15 locus display more rapid terminal differentiation and acquisition of effector function. However, these mice also exhibit more rapid loss of CD69 in the NK cell population. These data suggest that prolonged or enhanced NK cell activation in the absence of the m15 locus results in increased control of MCMV such that dissemination to the salivary gland is impaired.

### 2.7. MCMV-Δm15 Is Attenuated from First Seeding into the Salivary Glands and Shed at Reduced Levels into the Saliva

We have shown that the m15 locus targets NK cells to promote MCMV infection of the salivary glands. To characterise the attenuation of MCMV-Δm15 in this organ, we surveyed shedding of virus into the saliva of infected mice as a reflection of viral titres in the salivary glands. BALB/c and CBA mice were i.p. infected with 2 × 10^4^ PFU of MCMV and saliva samples collected on FTA paper from day 7 post-infection for quantification of viral DNA by RT-PCR. In susceptible BALB/c mice, both MCMV-Δm15 and MCMV-Rev are detected at similarly low levels from days 7–11 post-infection (Figure 7A). By day 14 post-infection, MCMV-Rev DNA is present at significantly higher levels in the saliva compared to MCMV-Δm15. From this timepoint onwards, while MCMV-Rev is shed at increasingly higher levels, reflecting amplification of MCMV in this organ, MCMV-Δm15 levels remain low in the saliva. In CBA mice, MCMV-Δm15 was completely absent from the saliva at all timepoints assessed, whereas MCMV-Rev could be detected from day 15 post-infection, with increased shedding until experimental endpoint (Figure 7B). Therefore, the role of the m15 locus on promoting viral dissemination to the salivary glands ultimately impacts viral shedding from the host.

We next sought to determine early viral titres in the salivary glands. Following clearance from peripheral organs, MCMV is disseminated via the blood to the salivary glands from five days post-infection [26,27]. As MCMV DNA levels in the saliva were low in the previous experiment, for these studies we infected mice with a higher dose of 2 × 10^5^ PFU MCMV and assayed viral titres in the salivary glands of BALB/c and CBA mice at seven days post-infection (Figure 7C,D). Even at these early timepoints, MCMV-Δm15 titres were significantly lower compared to MCMV-Rev in both strains of mice. This indicates that disruption of the m15 locus leads to defective seeding of the virus to the salivary glands. Indeed, MCMV-Δm15 DNA is undetectable in the blood of infected CBA mice, which is restored to wild type levels in the absence of NK cells (Appendix A). These data indicate that MCMV-Δm15 attenuation in the salivary glands is due to poor dissemination to the salivary glands.

## 3. Discussion

We have identified five 3′-coterminal transcripts produced by the previously uncharacterised region spanning the MCMV m14, m15 and m16 genes, which we have denoted the m15 locus. Disruption of all five transcripts led to attenuated MCMV replication in the salivary glands. This attenuation was rescued by NK cell, but not T cell depletion, indicating that the m15 locus plays a role in MCMV escape from NK cell mediated immunity. Infection with MCMV in the absence of the m15 locus led to enhanced NK cell activation and activity, which limits seeding of MCMV to the salivary glands and shedding into the saliva. This locus therefore actively promotes viral transmission via this route.

In this study we have shown that all five transcripts expressed by the m15 locus are required for mediating NK cell responses. Disruption of four transcripts or blocking translation from each of the annotated genes m14, m15 and m16, was insufficient. The m14–m16 region of MCMV has remained elusive in studies spanning several decades. In the first comprehensive characterisation of the m02 gene family, the expression of the m14, m15 and m16 proteins was noticeably absent in cellular localisation studies [6], although an m03 C-terminal HA tag protein was also undetected in this study, but was localised in a subsequent study using an N-terminal FLAG tag [28]. Notably, our own attempts to detect a C- or N-terminally tagged m15 protein consistently failed (data not shown). The expression of five transcripts across this site confounds attempts to produce C- and N-terminal tagged proteins. N-terminal tags to m14 and m15 would negatively impact several of the longer transcripts from this region. Likewise, as the transcripts we have detected over this locus do not correspond with the annotated genes, C-terminal tags may not be appropriately in frame to produce tagged proteins. Consequently, canonical gene production from this site is complicated, although T cells directed to an m14 product does indicate that protein is at least produced from the m14 gene [29]. Our transcript analysis provides an explanation for the difficulties encountered in other studies in identifying proteins from this region of the MCMV genome.

Previous reports of predicted or detected transcripts across the m15 locus have been largely inconsistent. Some global microarray or RNA-seq studies detected transcripts from all three genes [30,31], whereas others did not predict [32] or detect a transcript from the m14 gene [20]. A transcript spanning m15–m16, which corresponds with one of the five transcripts we detected in this study, and a spliced transcript over m15–m16 have also been described [20]. Another study involving RNA-seq comparison between MW97.01 (BAC-derived Smith) and Smith-GFP infected fibroblasts illustrated large differences in transcript profiles within the m02 gene family, especially over the m15 locus [33]. Few transcripts were detected over this region following MW97.01 infection, while multiple transcripts were identified in Smith-GFP infected cells. This was indicated to be due to several single nucleotide polymorphisms (SNPs) within m14 and possibly one within m15, highlighting that transcript profiles are likely to differ with genetic discrepancies. Our study is the first to characterise the m15 locus from the K181 strain of MCMV. Notably the sequences corresponding to the annotated m15 and m16 genes are highly variable between Smith and K181 strains [13], which may explain the differences in transcription profile between our study and others.

Transcription utilising alternative 5′-ends as shown here has also been described elsewhere in the MCMV genome [34,35,36] as well as within human CMV (HCMV) [21,37,38,39]. Tight temporal regulation of viral gene expression enables diversity of proteins produced during infection and the potential for targeted responses against host defences. This regulation dictates the outcome of infection in the host, as demonstrated in our study most markedly by the differential replication between MCMV-5′Δm15, with a deletion which only affects 4 transcripts, and MCMV-Δm15, in which all five transcripts are disrupted, despite both introducing frameshift mutations into m15. Similarly, wild type replication of MCMV-m14Stop, MCMV-m15Stop and MCMV-m16Stop indicate that the individual targeted genes alone do not affect MCMV replication in the salivary glands.

We have explored the possibility that the smallest transcript alone is responsible for the phenotype in MCMV-Δm15. However, this is unlikely to be due to protein produced from this transcript. There are two potential genes predicted over this transcript, the annotated m16 ORF and another ORF which extends 108 bp 5′ of m16, although the latter uses the rare near-cognate start site AAT. In our study, we have constructed two separate mutant MCMVs targeting the m16 gene and which would affect both of these genes as they are in-frame. The MCMV-m16Stop has a stop codon inserted after the first two methionines and the alternative MCMV-m16StopAlt has a stop codon inserted after the first four methionines. In the event of read-through, the next methionine is the 158th codon (of 210), which is outside of the region deleted in MCMV-Δm15 and therefore would not be responsible for its attenuated phenotype. Upon HCMV infection, mRNA has been detected from near-cognate start codons [21], although functionality has yet to be shown and near-cognate start codons are inefficient compared to AUG [40].

Potentially, the small transcript is a functional RNA. Approximately 500 viral miRNAs have been reported, with the majority expressed by herpesviruses including HCMV and MCMV [41]. However, miRNAs have specific motifs, none of which are predicted or have been detected experimentally within proximity of the m15 locus [31,42,43,44]. Recently, functional long non-coding RNAs (lncRNAs) have been described in all herpesvirus families [45,46,47]. Four major lncRNAs, representing more than half of the polyadenylated transcripts detected, have been described in HCMV, none of which overlap protein-coding genes [48,49]. While the small transcript we detect is not consistent with these features, we cannot rule out that it is a functional RNA.

The requirement for multiple genes to cooperate for a given phenotype is not without precedent. The m04 (gp34) protein binds and establishes MHC class I molecules on the surface of infected cells to regulate NK cell responses [50,51]. When present in mice expressing H-2D^k^, m04 leads to Ly49P dependent recognition of infected cells by NK cells, a process which also requires the MCMV MATp1 protein [52,53]. Deletion of m04 or MATp1 results in attenuated viral replication during acute infection, both of which are restored to wild type levels in the absence of NK cells [51,53]. The multifunctional m152 protein downregulates the NKG2D ligand RAE-1, as well as MHC class I and delays STING-mediated type I interferon signalling, effectively targeting three different arms of host immunity [54,55,56,57]. Downregulation of the cell surface expression of the RAE-1ε isoform, however, is incomplete in the presence of m152 alone and requires cooperation from m138 [58]. Single deletions of m138, which also downregulates stress-inducible ligands MULT-1 and H60, or m152 significantly attenuates acute MCMV replication [54,59]. Deletion of other NK immune evasion genes, such as m155 and m145 which target H60 and MULT-1, respectively, also leads to attenuated acute MCMV replication [60,61]. In all cases, the growth of these deletion viruses was restored following depletion of NK cells [54,59,60,61]. Notably, the phenotype of the m15 locus is unique in that while it targets NK cell responses, defective replication was not observed during acute infection, but rather later in the salivary glands, which was likewise rescued in the absence of NK cell depletion maintained throughout the infection. In the above examples of other MCMV genes cooperating, deletion of individual genes was sufficient for an observable phenotype; however, in our study, disruption of individual genes was insufficient and salivary gland attenuation required the disruption of all transcripts.

Disruption of the m15 locus led to heightened IFNγ secretion by and proliferation of NK cells. High expression of CD69 by NK cells was sustained 6.5 days after infection with MCMV-Δm15, suggesting that these cells continue to encounter antigen. While resident NK cells are functionally impaired and play a minimal role in the control of MCMV replication in the salivary glands [62], NK cells are important for limiting virus-induced secretory damage [63], potentially by controlling systemic inflammatory responses to acute infection. It is likely that the interaction between the m15 locus and NK cells during acute infection permits MCMV dissemination to the salivary glands. In support of this hypothesis, we have observed that MCMV-Δm15 is undetected in the blood of infected mice, but can be detected at wild type levels in mice depleted of NK cells. This complements the significantly lower titres of MCMV-Δm15 apparent from as early as day 7 post-infection in the salivary glands, consistent with the timing of early MCMV seeding into this organ.

A possible explanation for the maintenance of MCMV-Δm15 attenuation in the salivary glands, as evident by the monitoring of saliva shedding, is that in the absence of the m15 locus, continued defective viral dissemination results in insufficient infiltration of new virus into the salivary glands. Infected monocytes and/or dendritic cells (DC) travelling via the bloodstream are proposed to disseminate MCMV to the salivary glands, although the originating source of these infected cells is not clear, and can vary depending on the infection route [26,64,65,66]. While MCMV is cleared from visceral organs within a few days, lytic replication lingers in the salivary glands for weeks or months [67]. MCMV persists despite infiltration of virus-specific CD4+ T cells and the generation of a strong memory T cell response [68,69]. Farrell and colleagues have suggested that there is continual infection of DC from reactivated organs, which then seed MCMV to the salivary glands [66]. Our data suggests that in mice infected with MCMV-Δm15, there is a deficiency in the continued seeding of virus to the salivary glands, which holds viral loads at a constant low level in this organ. Nonetheless, inflating MCMV titres in the salivary glands are likely a combination of local viral proliferation and continual seeding to this organ. Therefore, at this stage, we cannot exclude the possibility that the m15 locus is important for both the dissemination of MCMV to the salivary glands and/or mediating site-specific replication in the salivary glands.

We conclude that contrary to conventional annotation, the m15 locus produces five 3′-coterminal transcripts spanning the predicted m14–m16 genes. Through early modulation of host immune responses, the m15 locus dictates the outcome of MCMV replication in the salivary glands, an important organ critical for amplification of virus in the host and a source of transmission in the population. Our study highlights the need for caution in the design and interpretation of future gene function studies. Deletion mutations targeted at a single gene may lead to false conclusions of the gene/s responsible for any phenotype observed. This study contributes to our understanding of the complexity of the expression potential of CMV genomes in maintaining the nuanced interaction with host immunity required to establish life-long infections.

## 4. Materials and Methods

### 4.1. Viruses and Cells

Primary mouse embryo fibroblasts (MEF) were cultured in minimal essential media (MEM) supplemented with neonatal calf serum (NCS). The bacterial artificial chromosome (BAC)-derived strain vARK25, referred to here as MCMV or wild type (WT) virus, has been previously shown to be biologically equivalent to K181 strain MCMV [70].

### 4.2. Mice

All experimental procedures in Australia were conducted in accordance with the NHMRC code of practice for the care and use of animals for scientific purposes, with approval by the University of Western Australia animal ethics committee (EAC RA/3/100/850 and RA/3/100/1274). Inbred mouse strains BALB/cArc (BALB/c; H-2^d^), CBA/CaHArc (CBA; H-2^k^) and SJL/JArc (SJL; H-2^s^) were purchased as specific pathogen-free (SPF) from the Animal Resources Centre (Murdoch, Western Australia) and maintained under minimal disease conditions. Experimental procedures undertaken in Croatia were approved by the Ethical Committee of the University of Rijeka and performed in accordance with Croatian Law for the Protection of Laboratory Animals that has been harmonised with the existing EU legislation (EC Directive 86/609 EEC). Mice were bred and housed under SPF conditions.

### 4.3. BAC Mutagenesis

All BAC mutagenesis was performed with the K181 BAC, pARK25 [70] and all genomic locations correspond to the published K181 sequence [13] (accession # AM886412).

The deletion mutants MCMV-Δm14–16 and MCMV-Δm15 were generated by ET recombination as previously described [70]. Briefly, the kanamycin resistance gene (Kn^R^) flanked by FRT sites was PCR amplified from pSLFRTKn and inserted into pARK25 via red-mediated recombination between 49 bp of flanking sequences homologous to the target region in pARK25. The Kn^R^ cassette was then excised via flp-mediated site-specific recombination between flanking FRT sites, leaving a remnant FRT site over the region of interest. In MCMV-Δm14–16, 2489 bp of sequence encompassing m14, m15 and m16 was deleted (nt 13, 134 to 15, 622). MCMV-Δm15 has a concomitant deletion of 110 bp (nt 14, 868 to 14,977) at the 3′-end of the m15 gene. To construct the revertant of MCMV-Δm15, denoted MCMV-Rev, a 4006 bp fragment (nt 12,417 to 16,422) of MCMV flanked by AttB Gateway cloning sites was subcloned into a constructed Gateway cloning vector, pST76K-SR-DEST, based on the shuttle vector pST76K-SR [71]. The 4 kb fragment includes the 110 bp of wild type MCMV sequence to be restored flanked by 2 kb of homologous MCMV sequences to the integration site. pST76K-SR-DEST-m15 was used to seamlessly reintroduce full length m15 into MCMV-Δm15 by two-step replacement between the 2 kb of flanking homologous MCMV sequences to generate MCMV-Rev.

Recombinant viruses MCMV-m14stop, MCMV-m15stop, MCMV-m16stop and MCMV-5′Δm15 were constructed using seamless two-step ET recombination. In the first instance a dual selection cassette (CcdB/Kn^R^) was constructed to encode the Kn^R^ for positive (first step) selection and the CcdB cassette for (second step) counter selection. This plasmid, denoted pInCK, contains the CcdB cassette from pDONR201 (Invitrogen, Waltham, MA, USA) directionally inserted downstream of the arabinose inducible promoter in pBAD30 and adjacent to Kn^R^ from pSLFRTKn on the pGEM-T backbone (Promega, Madison, WI, USA). To introduce a premature stop codon into the m14 gene, PCR primers were designed to amplify the CcdB/Kn^R^ cassette flanked by homologous MCMV sequences. The resultant PCR product was inserted into pARK25 propagated in DY380, allowing heat inducible recombinase functions. Successful insertion of the dual selection cassette over the region of interest via flanking homologous MCMV sequences was verified by bacterial growth in the presence of 25 μg/mL Kn, 15 μg/mL chloramphenicol (Cm) and 0.2% D-glucose, the latter of which suppresses CcdB expression. The CcdB/Kn^R^ cassette was excised via a second ET recombination event with a SOE PCR generated linear DNA fragment containing a guanidine to thymidine substitution at nt 13,135, which alters the 18th codon of m14 from GAA (glutamine) to TAA (stop) and flanked by homologous MCMV sequences. Successful removal of the CcdB/Kn cassette was screened by assessing CcdB sensitivity in the presence of 15 μg/mL Cm and 0.2% L-arabinose.

For construction of MCMV-m15stop, the CcdB/Kn cassette was amplified from pInCK and inserted into the m15 gene via seamless ET recombination as described above. A thymidine to adenine substitution at nt 14,113, which alters the 7th codon of the m15 gene from TAT (tyrosine) to TAA (stop), was introduced into a linear DNA fragment by SOE PCR and substituted into pARK25 as described for MCMV-m14stop to give MCMV-m15stop. For construction of MCMV-m16stop, the CcdB/Kn cassette was amplified from pInCK and inserted into the m16 gene via seamless recombination as described above. A cytosine to adenine substitution at nt 15,087, changing the 16th codon from TAC (tyrosine) to TAA (stop) was introduced into a linear DNA fragment by SOE PCR and substituted into pARK25 as described to give MCMV-m16stop. An alternative recombinant MCMV targeting m16, MCMV-m16stopAlt, had a thymine to adenine substitution introduced at nt 15,125, changing the 29th codon of m16 from TTA (Leucine) to TAA (stop) and inserted into pARK25 as described for MCMV-m16stop. This mutation is downstream of four potential methionine start sites in m16.

To produce MCMV-5′Δm15, the CcdB/Kn cassette from pInCK was inserted into pARK25 as for MCMV-m15stop by seamless ET recombination. A 358 bp deletion (nt 14,154 to 14,518) was introduced into a linear DNA fragment by SOE PCR and substituted into pARK25 to give MCMV-5′Δm15. All recombinant viruses were verified by RFLP analysis and Sanger sequencing of the target gene. Primers for BAC mutagenesis are listed in Appendix A.

### 4.4. Rescue of Recombinant Viruses

BAC DNA was extracted using the NucleoBond^®^ Xtra Midi kit (Macherey-Nagel, Duren, Germany). Recombinant MCMVs were rescued by transfection of viral BAC DNA into MEF as previously described [70]. Recombinant viruses were propagated on MEF to generate viral stocks. Normal in vitro replication of all recombinant viruses was confirmed by multi-step growth curve analyses as described previously [70].

### 4.5. Pathogenesis in Mice

For all pathogenesis experiments, 6–8 week old female mice were used. For acute experiments, BALB/c mice were inoculated via the intraperitoneal (i.p.) route with 1 × 10^5^ plaque forming units (PFU) of tissue culture-derived virus (TCV). CBA mice were inoculated with 1 × 10^4^ PFU of salivary gland-derived virus (SGV) generated in 3 week old CBA mice. SJL mice were inoculated with 1 × 10^4^ PFU of SGV generated in 3 week old SJL mice. Spleen and liver were extracted at day 3 post-infection to determine viral load. For chronic infection, all mice were i.p. inoculated with 2 × 10^4^ PFU of TCV. Salivary glands were extracted at day 18 post-infection. All organs were flash frozen and stored at −80 °C. Organs were individually weighed, homogenised in cold MEM supplemented with 2% NCS at 100 mg of tissue/mL and clarified at 850× *g* for 10 min at 4 °C. Viral titres were quantified on MEF by standard plaque assay.

### 4.6. Depletion of Lymphocyte Subsets

For total T cell depletion, CBA mice were i.p. injected with a combined dose of 100 μg of anti-CD4 Ab GK1.5 and 100 μg anti-CD8 Ab YTS169 (Perkins Monoclonal Antibody Facility, Perth, Australia) on days −3, −1, 0 (day of infection) and every fifth day thereafter until experimental endpoint at day 18 post-infection. Successful depletion was monitored with by flow cytometry using antibodies for CD4 specific FITC RM4-4 (eBioscience, San Diego, CA, USA) and CD8 specific PE 53-6.7 (BD). Control animals were treated with either PBS or 200 μg of isotype control GL121 (Perkins Monoclonal Antibody Facility) on indicated days. For NK cell depletion, CBA mice were i.p. injected with 200 μg of anti-asialo GM1 (Wako Chemicals, Osaka, Japan) on days −1, 0 and every fourth day thereafter until experimental endpoint at day 18 post-infection. Successful depletion was monitored by flow cytometry using antibodies for anti-CD3ε (eBioscience) and anti-NKp46 (eBioscience). Control animals were left untreated or injected with PBS on treatment days.

### 4.7. Saliva and Blood Collection for Quantification of MCMV DNA by RT-PCR

MCMV shed into saliva was collected on FTA filter paper (Whatman, Maidstone, UK) and prepared for RT-PCR as described previously [72]. To extract viral DNA from the bloodstream of infected mice, blood was collected from infected mice by tail vein bleed into a microtainer tube lined with K_2_EDTA (BD). PBS was added to 50 μL of blood to a total volume of 1 mL, mixed by inversion and incubated at 37 °C for 30 min to settle out red blood cells. The buffy coat was transferred to a fresh microfuge tube and viral DNA was extracted using a Wizard SV Genomic DNA kit (Promega, Madison, USA) according to manufacturer’s instructions. Quantification of viral DNA was conducted using the LightCycler 480 (LC480) real-time PCR system (Roche Diagnostics, Basel, Switzerland) with M86 specific primers (forward 5′-CCGTGTCCACCAGTTTGATCTTGT-3′; reverse: 5′-TGGACAACCCGGAGACCTACAC-3′) and a gene-specific oligonucleotide probe (FAM–CTGGTCCAGGAACTGCGACT-BHQ-1; Geneworks Pty Ltd., Thebarton, Australia). Each RT-PCR reaction contained 10 μL 2× Probes Master Mix (Roche), 5 pmol of each primer, 5 pmol of probe and 1 mm^2^ of saliva impregnated FTA paper or 4 μL of total DNA extracted from the blood. Cycling conditions were as follows: 95 °C for 10 min, 40× (95 °C for 30 s, 66 °C for 120 s, 50 °C for 10 s). Viral DNA concentration was calculated relative to an internal standard control using the fit-points method from the LC480 software.

### 4.8. Isolation and Phenotypic Analysis of NK Cells by Flow Cytometry

For flow cytometric analysis, CBA mice were i.p. infected with 2 × 10^5^ PFU of TCV MCMV. Splenocytes were extracted and prepared as previously described [73] and incubated with 2.4G2 monoclonal antibody to reduce nonspecific staining. The following monoclonal antibodies were used for cell surface staining: anti-CD3ε (eBioscience), anti-NKp46 (eBioscience), anti-CD11b (eBioscience), CD27 (BD), anti-CD69 (eBioscience).

To assess NK cell proliferation, CBA mice were i.p. injected with 2 mg of BrdU (Sigma) and splenocytes extracted 2 h later. To detect BrdU incorporation, prepared splenocytes were first stained for surface antigens prior to fixation, permeabilisation, refixation, treatment with DNase I and intracellular stained according to manufacturer’s instructions (BrdU flow kit; BD, Franklin Lakes, USA).

To detect IFNγ production by NK cells, splenocytes were incubated in RPMI supplemented with 10% fetal calf serum (FCS) for 5 h in the presence of 500 IU/mL of IL-2 and 1 μg/mL of brefeldin A (BFA; eBioscience) at 37 °C. Cells were surface stained, then fixed and permeabilised using Cytofix/Cytoperm solutions (BD Pharmingen, San Diego, CA, USA) followed by intracellular staining according to manufacturer’s instructions.

### 4.9. 5′-RACE and 3′-RACE PCR

MEF were seeded at 1 × 10^6^ per well of a 6-well tray 24 h prior to infection with MCMV at a multiplicity of infection (MOI) of 1 in the presence of 250 μg/mL phosphonoacetic acid, which was maintained throughout the experiment. Virus was aspirated after 1 h incubation at 37 °C. Cell monolayers were washed with citric acid buffer for 1 min to remove bound virus, followed by neutralisation with MEM supplemented with 2% NCS for 5 min. Total RNA was extracted using the PureLink RNA mini kit (Invitrogen) at 24 h post-infection for 3′- and 5′-rapid amplification of cDNA ends (RACE). Poly(A)+-selected RNA was prepared using the MicroPoly(A)Purist kit (Ambion, Austin, TX, USA). To map the 5′- and 3′-ends of transcripts, 5′- and 3′-RACE PCR was conducted on poly(A) + RNA and total RNA, respectively, according to manufacturer’s instructions from the FirstChoice^®^ RLM-RACE kit (Ambion). MCMV specific primer sequences used for primary and nested 5′- and 3′-RACE PCR are available upon request. Both 5′- and 3′-RACE PCR products were gel-purified and Sanger-sequenced.

### 4.10. Sequencing

DNA samples were gel purified and submitted for sequencing on the SOLiD4 platform at the LotteryWest State Biomedical Facility Genomics located at Royal Perth Hospital, Perth, Western Australia, Australia. Analysis of sequences, construction of recombinant viral MCMV genomes and sequence alignments were conducted using CLC Main Workbench (CLC bio A/S).

### 4.11. Statistical Analysis

All statistical analyses were performed using GraphPad Prism 5.0 (GraphPad Software Inc., San Diego, CA, USA). Statistical significance between two groups was determined using the unpaired two-tailed Mann–Whitney test. Statistical significance between three or more groups was determined using one-way analysis of variance (ANOVA) followed by Tukey’s comparison, except where there was only one value repeated within a sample group and then one-way ANOVA on ranks followed by Dunn’s multiple comparison test was used. All data are shown as mean ± standard error of the mean (s.e.m).

## Figures and Tables

**Figure 1 pathogens-10-00866-f001:**
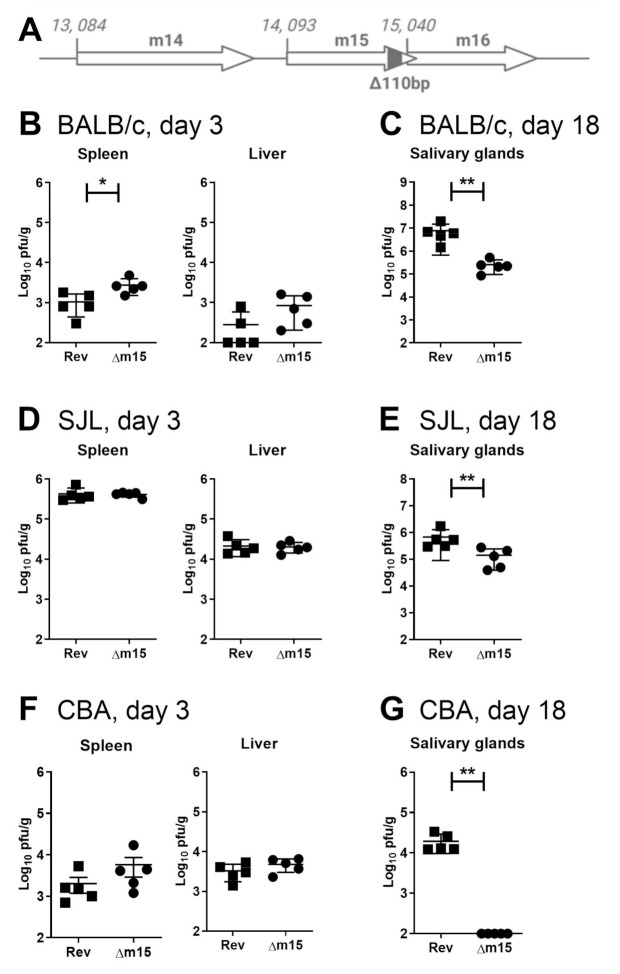
Deletion within the m15 gene leads to defective MCMV replication in multiple mouse strains. (**A**) Schematic representation of the 110 bp deletion from the 3′-terminus of the m15 gene. (**B**) BALB/c mice were infected i.p. with 1 × 10^5^ PFU of tissue culture-derived virus and (**D**) SJL and (**F**) CBA mice infected with 1 × 10^4^ PFU of salivary gland-derived virus to assess viral titres in the spleen and liver at day 3 post-infection. (**C**,**E**,**G**) All mice were infected with 2 × 10^4^ PFU of tissue culture-derived MCMV for measurement of viral titres in the salivary glands at day 18 post-infection. Error bar depicts s.e.m. and horizontal bar denotes the mean. Asterisks denote statistical significance (unpaired two-tailed Mann–Whitney test): * *p* < 0.05; ** *p* < 0.01. The x-axis is set at the detection limit.

**Figure 2 pathogens-10-00866-f002:**
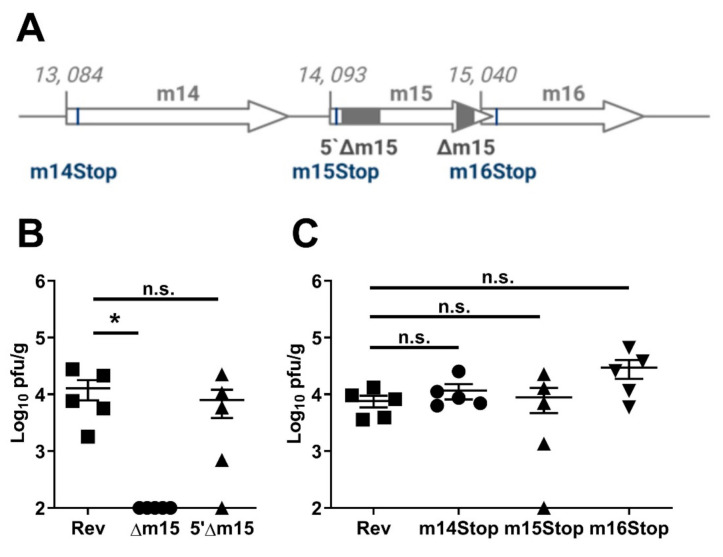
Deletion at the 5′-terminus of m15 and individual genes does not affect MCMV replication. (**A**) Schematic representation of the five recombinant MCMVs targeting the m14, m15 and m16 genes. 5′Δm15 has 358 bp and Δm15 has 110 bp deleted from the 5′- and 3′-terminus of m15, respectively (grey boxes). A single nucleotide substitution within the m14, m15 and m16 genes introduces a premature stop codon in m14Stop, m15Stop and m16Stop (black lines). (**B**) CBA mice were i.p. infected with 2 × 10^4^ PFU of MCMV and salivary glands extracted at day 18 post-infection for assaying viral titres. 5′Δm15 replicated as well as wild type MCMV, whereas Δm15 was attenuated in the salivary glands. (**C**) Insertion of a stop cassette at the start of the m14 (m14Stop), m15 (m15Stop) and m16 (m16Stop) genes did not affect MCMV replication in the salivary glands. Error bar depicts s.e.m. and horizontal bar denotes the mean. Asterisk denotes statistical significance (one-way ANOVA): * *p* < 0.05; n.s., not significant. The x-axis is set at the detection limit.

**Figure 3 pathogens-10-00866-f003:**
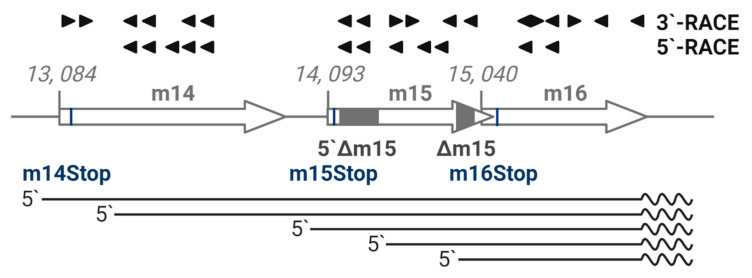
Multiple 3′-coterminal transcripts are produced by the m15 locus. RNA was extracted from primary MEF infected with MCMV (MOI = 1) in the presence of phosphonoacetic acid. 5′- and 3′-RACE PCR was conducted on poly(A)+ RNA and total RNA, respectively, with indicated primers denoted by arrowheads spanning the m14–m16 region. Sanger sequencing revealed five 3′-coterminal transcripts. Transcripts detected by RACE analysis are depicted by straight lines, polyA tails are depicted by the wavy lines.

**Figure 4 pathogens-10-00866-f004:**
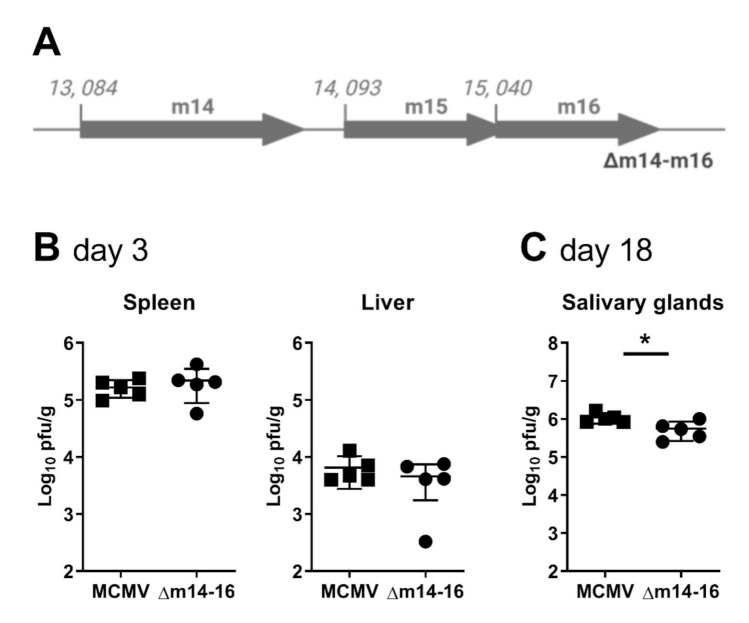
Deletion of the m15 locus attenuates MCMV replication in the salivary glands. (**A**) Schematic representation of recombinant MCMV with the entire m15 locus deleted (shown as greyed out arrows). BALB/c mice were i.p. infected with (**B**) 1 × 10^5^ PFU for acute infection or (**C**) 2 × 10^4^ PFU for chronic infection. Viral titres in specified target organs were assessed by standard plaque assay at (**B**) 3 days and (**C**) 18 days post-infection. Error bar depicts s.e.m. and horizontal bar denotes the mean. Asterisks denote statistical significance (unpaired two-tailed Mann–Whitney test): * *p* < 0.05. The x-axis is set at the detection limit.

**Figure 5 pathogens-10-00866-f005:**
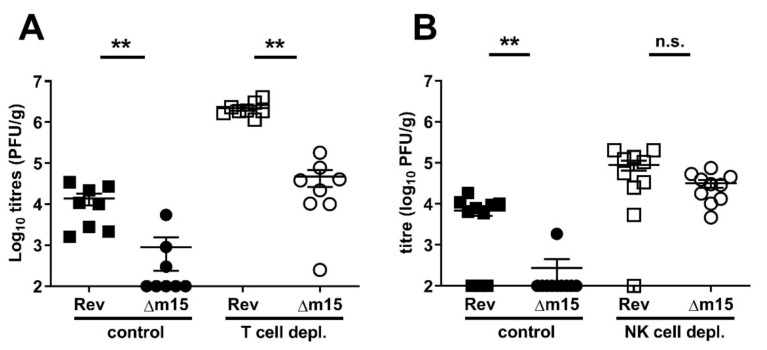
NK cells, but not T cells, are targeted by the m15 locus. (**A**) CBA mice were depleted of total T cells by combined treatment with anti-CD4 GK1.5 and anti-CD8 YTS169 at −3, −1, 0, 5, 10, 15 days post-infection. Control mice were treated with PBS only or isotype control GL121. Mice were i.p. infected with 2 × 10^4^ PFU MCMV and salivary gland titres assessed at day 18 post-infection. Δm15 remains attenuated in the salivary glands in the absence of T cells. (**B**) For depletion of NK cells, CBA mice were treated with anti-asialo-GM1 at −1, 0, 4, 8, 12, 16 days post-infection. Control mice were treated with PBS. Mice were i.p. infected with 2 × 10^4^ PFU Rev or Δm15 and salivary gland titres assessed at day 18 post-infection. Δm15 titres were restored to wild type levels in mice depleted of NK cells. Data are combined from two independent experiments. Error bar depicts s.e.m. and horizontal bar denotes the mean. Asterisks denote statistical significance (unpaired two-tailed Mann–Whitney test): ** *p* < 0.01; n.s., not significant. The x-axis is set at the detection limit.

**Figure 6 pathogens-10-00866-f006:**
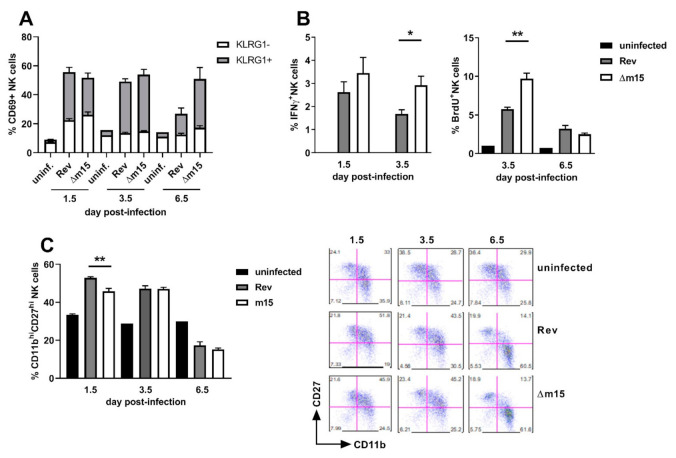
Disruption of the m15 locus enhances NK cell responses during acute MCMV infection. CBA mice were left uninfected or i.p. infected with 2 × 10^5^ PFU of MCMV. Splenocytes were extracted at indicated timepoints and surface stained with NKp46 and CD3ε to identify NK cells. (**A**) Cells were stained for the expression of activation marker CD69 and terminal exhaustion marker KLRG1. Bars show the percentage of cells expressing CD69, of those that are KLRG1- and KLRG1+ are shown in white and grey, respectively. (**B**) NK cells were stained for IFNγ production and BrdU incorporation by proliferating NK cells. (**C**) NK cells were stained for the expression of maturation markers CD27 and CD11b with representative dot plots shown. Error bar depicts s.e.m. Asterisks denote statistical significance (one-way ANOVA): * *p* < 0.05; ** *p* < 0.01.

**Figure 7 pathogens-10-00866-f007:**
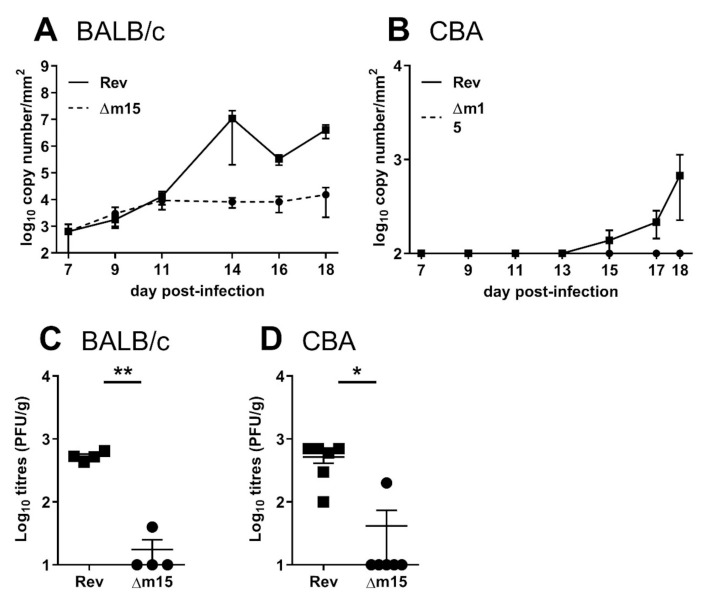
The m15 locus is required for first seeding of MCMV into the salivary gland and shedding into the saliva. (**A**) BALB/c and (**B**) CBA mice were i.p. infected with 2 × 10^4^ PFU of MCMV and saliva collected on FTA paper at indicated timepoints. MCMV DNA in the saliva was quantified by M86 RT-PCR. Data from at least five mice are shown per timepoint. (**C**) BALB/c and (**D**) CBA mice were i.p. infected with 2 × 10^5^ PFU of MCMV and viral titres in the salivary glands assessed at day 7 post-infection. The x-axis is set at the limit of detection. Error bar depicts SEM and horizontal bar denotes mean. Asterisks denote statistical significance (unpaired two-tailed Mann–Whitney test): * *p* < 0.05; ** *p* < 0.01. The x-axis is set at the detection limit.

## Data Availability

No publicly archived data available.

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
