# Peer review of "The m15 Locus of Murine Cytomegalovirus Modulates Natural Killer Cell Responses to Promote Dissemination to the Salivary Glands and Viral Shedding"

_pathogens, 2021, doi:10.3390/pathogens10070866_

Round 1

Reviewer 1 Report

This is an interesting study, which demonstrates some strong in vivo phenotypes for viruses deleted of a genetic region that has not been previously assigned a function. The authors convincingly show that deleting the m15 locus has a strong NK-dependent effect on seeding and/or replication of MCMV in the salivary gland of mice.

Having said that, I do have some comments about their conclusions regarding the mechanistic elements of the work. My biggest concern is their conclusion that the entire region, encoding 5 transcripts, is required for the phenotype they observe. Their data clearly shows that deletion of 110bp from the 3’ end of m15 has the same phenotype as deletion of the entire region. While this will obviously cause a 110bp deletion in all five transcripts, this doesn’t mean that all 5 are required. Until more is known about exactly what proteins are produced from this region, it is premature to conclude that all proteins in the region must co-operate. The data is entirely consistent with just the shortest transcript being required.

Can the authors perform a bioinformatic analysis of the transcripts, to suggest which proteins might be expressed from this region? They also need to mention the possibility of their stop codon mutants suffering from read-through and/or use of a later ATG site, and could comment that the effect does not have to be due to a protein, there are plenty of examples of functional RNAs.

In addition, the authors need to be careful about attributing cause & effect to the m15 gene region (e.g. line 312). They have some striking NK-dependent phenotypes, but they don’t present any data to demonstrate that these effects are due to a direct impact of the m15 gene region on NK cells – that would require reductive in vitro experiments. Similarly, line 346 they claim that this gene region is a modulator of shedding – they can’t claim this is a direct function, it’s likely an indirect consequence of impacts on dissemination.

Minor comments:

Line 133. The authors state that their original hypothesis of strain-dependent effects was wrong, yet the different mice strains do have different phenotypes?

Fig 4C – the effects seem very small here. Why did they not use CBA mice for this?

Line 247 – can the authors show the fold-change between the viruses following T-cell depletion (and controls), to demonstrate that the magnitude of the phenotype remains the same in the presence/absence of T-cells?

Line 298 – the authors claim this higher levels of NK activation is due to higher levels of spleen replication, but CBA mice don’t have higher levels of virus in the spleen (Fig 1).

Line 322 – not all effects were at d1.5. CD69 was d6.5, IFNg & BrdU were at d3.5.

Methods – can the authors give more detail on virus construction, it’s impossible to work out exactly what was done from the descriptions given. E.g. Line 510, I assume these cassettes were resolved to provide seamless mutation? What was the FRT site for? Was the re-insertion of m15 in Rev seamless, or were sites such as Gateway included?

The authors frequently comment that they used the Revertant as the control, because it controls for any other mutations introduced during construction. Yet presumably the revertant is only a revertant of one of the mutants, so the others could contain unintended errors? The easiest thing to do would simply be to NGS their viruses to make there are no errors?

Stats – the authors use T-tests, yet for most of their assays, they have multiple samples, in which case a one- or two- way ANOVA is more appropriate.

Reviewer 2 Report

In this manuscript, the authors identified five novel transcripts spanning the MCMV m14, m15 and m16 genes (designated the m15 locus). Disruption of any of these genes in this region (m14, m15, or m16) did not affect viral replication in vivo. However, inactivation of all of these five transcripts strikingly attenuated viral replication in the salivary glands of several strains of laboratory mice. Interestingly, depletion of NK cells, but not T cells, restored viral replication in salivary glands. Further, the authors observed that disruption of the m15 locus resulted in the elevated NK cell responses, as suggested by the increased NK cell proliferation and IFN-γ production. The results are clear and well presented. In general, this manuscript is well written and presents an interesting story that contributes to better understanding of transcription complexity of the CMV genomes. The specific comments are described below.

  1. Did you validate these five novel transcripts by northern blots? What are the expression abundances for them?
  2. In lines 222-224, the authors state that “More importantly, like MCMV-∆m15, MCMV-∆m14-16 reached significantly lower titres compared to wild type MCMV in the salivary glands of BALB/c mice at day 18 post infection (Figure 4C).” I don’t think this description is appropriate because the replication difference between MCMV-∆m14-16 and WT are very small when you compared to the data Fig. 1C, which shows more than 10-fold difference between MCMV-∆m15 and Rev viruses.
  3. In the manuscript, the authors saw the most sticking phenotype in CBA mice. Did you test MCMV-∆m14-16 mutant in these mice? You probably could see bigger phenotypes in CBA mice as compared to that in BALB/c mice as you showed in Fig. 4C.

Round 2

Reviewer 1 Report

The authors have addressed all my concerns.